# Nurses and Stigma at the Time of COVID-19: A Phenomenological Study

**DOI:** 10.3390/healthcare10010025

**Published:** 2021-12-24

**Authors:** Silvio Simeone, Teresa Rea, Assunta Guillari, Ercole Vellone, Rosaria Alvaro, Gianluca Pucciarelli

**Affiliations:** 1Department of Experimental Medicine, University of Campania Luigi Vanvitelli, 80138 Naples, Italy; silviocecilia@libero.it; 2Department of Public Health, University of Naples “Federico II”, 80131 Naples, Italy; teresa.rea@unina.it; 3Department of Biomedicine and Prevention, University of Rome “Tor Vergata”, 00133 Rome, Italy; ercole.vellone@uniroma2.it (E.V.); rosaria.alvaro@gmail.com (R.A.); g.pucciarelli81@gmail.com (G.P.)

**Keywords:** stigma, nurse, COVID-19, lived experience, phenomenological

## Abstract

The COVID-19 pandemic is putting strain on every country in the world and their health systems. Healthcare professionals struggle on the frontline and they can experience stigma, which can create difficulties in controlling epidemic diseases, influencing the mental health of healthcare professionals, caregivers, families, communities, and the provided quality of care. The aim of this study is to explore the lived experience of Italian nurses about perceived stigma during COVID-19 pandemic with the phenomenological Cohen method. The principal themes that emerged from data analysis were “stigma in the working environment” and “stigma in everyday life”. Each of these themes had subthemes: “looks like gun sights”, “avoiding closeness to others”, “nobody wants to touch you”, and “the fault of being your family members”. Public health emergencies, such as the COVID-19 pandemic, are stressful events for individuals and communities. Stigma can be more dangerous than the disease, and a major obstacle to appropriate medical and mental health interventions. Understanding how healthcare professionals experience stigma is essential to design and implement specific educational, psychological, and organisational programmes.

## 1. Introduction

In December 2019, the first cluster of a new pneumonia caused by SARS-CoV-2 appeared in China, which causes coronavirus disease 19 (COVID-19) [1,2]. The virus spread quickly, evolving from a health emergency of international concern [3] to a global pandemic [4,5]. The United States and the European continent have experienced the highest number of infections and deaths [6], although a precise estimate is difficult because data change rapidly [7].

The main source of COVID-19 is human-to-human transmission [8], and it can occur in both symptomatic and asymptomatic subjects [6]. The main symptoms reported in the literature range from mild fever, fatigue, and gastrointestinal problems reported in mildly symptomatic patients to dyspnoea with high fever, pneumonia with marked hypoxia, acute respiratory distress syndrome, heart problems, and severe organ failure in patients with more severe symptoms [6,9,10]. Older adults face greater risk in the event of contracting COVID-19 [11], as do those affected by cardiovascular diseases, hypertension, and diabetes [12].

Healthcare professionals work on the frontlines during epidemics of highly infectious diseases [13]; therefore, they are at greater risk of spreading the virus, even outside their working context [14]. This has important repercussions on the mental health of workers [15,16]. Nurses are the health professionals most in contact with patients and the general population. As described during other health emergencies, during COVID-19, nurses have an increased risk of becoming infected, even going so far as to risk their lives in service [8,17]. In addition to being more at risk of contracting COVID-19, healthcare professionals may be at risk of developing mental health problems because of COVID-19, such as anxiety, depression, and hostility [18,19]. In addition to this, healthcare professionals who manage these situations risk being stigmatised by the general population. Fear and anxiety about a disease can lead to social stigma towards people, places, or things [14,20,21]. Stigma is a powerful social process that is characterised by labelling, stereotyping, and separation, leading to status loss and discrimination, all occurring in the context of power [22]. The experience of stigmatisation can have a negative effect on people affected by the disease [23,24], and on their caregivers [25], families [26,27], friends, and communities (CDC, 2020).

During previous pandemics, healthcare professionals have been victims of stigma. During the SARS epidemic in Taiwan, about 20% of involved healthcare professionals experienced stigma and rejection by neighbours [28] because of their profession [29]. In the MERS epidemic [30], there were prejudice, stigma, and social isolation directed towards the medical staff. This was associated with a reduction in participants’ self-esteem, and an alteration to the meaning of their existence and their perception of belonging to a community.

Although several studies analysed stigmatisation towards health personnel in previous pandemics [14,27,28,29,30], to our knowledge, few studies have analysed the perception of stigma against nurses during COVID-19. This represents a gap in the literature, as studies on COVID-19 have only focused on the treatment of patients, often neglecting the impact of the pandemic on the health workforce. As specified by the WHO, stigmatisation during the current COVID-19 pandemic could create a situation in which the virus has a greater probability of spreading. This could lead to more serious health problems and major difficulty in controlling the pandemic [31,32].

Therefore, the aim of this study is to explore the lived experience of nurses about perceptions of being victims of stigmatisation.

## 2. Material and Methods

### 2.1. Design

We used Cohen’s phenomenological methodology [33] to conduct this study, which combines characteristics of descriptive (Husserlian) and interpretative (Gadamerian) phenomenology.

Descriptive phenomenology requires the suspension of all ideas and suppositions in order to access the nature of consciousness itself; thus, it is possible to obtain the meaning of the individual’s experience. In descriptive phenomenology, daily conscious experiences must be described, while any preconceived opinions must be set aside and placed in parentheses [34].

Interpretative phenomenology is an approach that aims to provide detailed examinations of personal lived experiences. It is a particularly useful methodology for examining complex, ambiguous, and emotionally charged topics [35].

Cohen’s phenomenology [33] is divided into the following phases. 

(1)Bracketing: critical reflection technique that precedes data collection. Researchers are asked to write their own beliefs and hypotheses on the phenomenon to avoid contamination of the analysis by prejudices on the subject.(2)Interview of the participants with open questions. During the interviews, the researcher takes notes in the field, noting nonverbal language, describing the environment and any noteworthy detail. Data collection and analysis are iterative processes that are simultaneously conducted. The interviews continue until the data are saturated (the point at which new collected and analysed data no longer bring additional insights to the research questions).(3)Interviews are transcribed verbatim and checked for accuracy.(4)Investigators immerse themselves in the data by repeatedly reading and rereading interview transcripts and field notes.(5)Extrapolation of the main themes and any subthemes.(6)Extrapolated themes are compared among all the researchers involved, discussing and determining their reliability.(7)Extracted themes are presented to the participants of the study participants for their confirmation.(8)The scientific report is produced.

### 2.2. Tested Group

During the COVID-19 pandemic in Italy, in March and April 2020, we enrolled an intentional sample of participants through sampling a nonprobabilistic “snowball”. Regardless of facility or region of origin, the contacted subjects to participate in the study had to meet the criteria: (1) nurses working in COVID-19 hospitals (publicly recognised as hospitals caring for COVID-19 patients), (2) worked continuously during the highest COVID-19 alert period in Italy, (3) had experienced one or more episodes of stigmatisation attributable to their work in the last period, (4) consented to participate in the study and signed the informed consent form. Exclusion criteria were (1) not being in continuous service during the months of sampling and (2) refusing to participate in the study. Each participant was free to refuse or withdraw from the study at any time.

### 2.3. Data Collection

In line with the chosen methodology, before data collection, each investigator involved in the study performed bracketing, putting aside their ideas and prejudices about study. Bracketing reduces the possibility that researchers introduce their ideas rather than seeing the data from the participants’ perspective [33].

The participants had no previous contact with the researchers. Interviews were conducted by Internet app at home without anyone else being present. The interviews were conducted using a single open question; this granted participants maximal freedom of expression. In this way, the participants’ “world” became the focus of the research [36].

Before asking the participants to describe their experience of any episodes of suffered stigmatisation, the definition of stigma was explained: the etymological meaning is “brand”, meaning the attribution of negative characteristics that differentiate the individual from the norm. It is a social label that could generate behaviours that are not normally found.

During the interviews, researchers that carried out the interviews (S.S., A.G., and E.V.) had a “welcoming attitude” [37,38], expressing cordiality and reassurance, and not judging participants’ narratives to facilitate their description of the experience. During the interviews, the researchers wrote field notes: personal reflections relating to the setting and the nonverbal language used by the interviewees in a diary.

Data saturation, namely, the redundancy of the extracted themes [36], was achieved after 19 interviews. All interviews were audio recorded and lasted between 25 and 40 min. To allow for researchers to become familiar with the interviews and the participants, two interviews had been conducted that were not included in the analysed sample. To comply with national provisions to contain the spread of COVID-19, issued to protect the health of Italian citizens, this study used of voice over Internet protocol technology. The Zoom app, a cloud-based video conferencing service that allows for communicating in real time with geographically distant people via computer, tablet, or mobile device, was used. This app was identified as the most useful in qualitative research [39,40]. These technologies replicate the characteristics of traditional face-to-face interviews, including the ability to perceive and observe nonverbal language [41].

### 2.4. Data Analysis

Data analysis was performed by each researcher on each individual interview. Interviews were transcribed verbatim and supplemented with field notes. Each transcript was read several times to first obtain an overview of the participants’ experiences. Subsequently, each researcher reread each transcription of each individual interview line by line to determine the meaning of the experiences. The extrapolated themes by the researchers from the participants’ experiences were compared among all researchers, with no disagreements arising. Researchers then proceeded to confirm the extrapolated themes by proposing the same themes to the participants. Meeting the Lincoln and Guba [42] criteria for qualitative research further ensured the scientific rigour of this study.

The interviews, data analysis, and verification of the results were conducted in Italian. After having prepared the scientific report, translation processes and back translation were performed according to the WHO methodology for the validation of instruments in different cultures and languages from the source language [43]. This methodology focuses on conceptual content rather than the literal equivalent. It was thus possible to ensure that the meaning behind the obtained data was respected.

### 2.5. Ethical Aspect

The institutional review board of the University of Rome Tor Vergata approved the study. The study respects the Helsinki Declaration principles. Anonymity was guaranteed by coding participants’ identities.

## 3. Results

We examined 19 nurses (Table 1) with an average age of 51.57 years (DS, 8.9; range 25–60) and a median of 44 years. More than half of our sample was married or had a stable partner and children.

All the interviewed subjects worked in public hospitals, about 70% in a region of Southern Italy, and 15% worked in central Italy.

About 50% of our sample did not work in specific COVID-19 units and had no positive cases within their work unit.

From data analysis, we identified the principal theme of stigma in the working environment, and the two subthemes of “gun-sight looks” and “avoiding closeness to others”. Another principal theme was “stigma in everyday life”, with the two subthemes of “nobody wants to touch you” and “the fault of being your family members”.

### 3.1. Stigma in the Working Environment

Participants described episodes related to stigma within their work context. The workplace for these health professionals represented a sort of second home. Participants described well their sense of isolation within a context that they had always considered their own—a family context.

Some participants experienced clear episodes of stigma, while for others, such episodes were less evident. Echoing the pattern of Earnshaw and Chaudoir’s Health Stigma Framework [44], also used in other contexts [45,46], we categorised the stigma in its various forms and interpretations as either lived or perceived. That is, there are clear discriminating episodes of stigmatisation and episodes evaluated as such, internalised by stigmatised subjects. Our results show that some attitudes were considered to be stigmatising as a result of reflection and not immediately, as occurred for obvious behaviours. The subtheme “gun-sight looks” clearly represents internalised stigma, while “avoiding closeness to others” represents clearly perceived stigma behaviours.

I never thought there could be a definition for what I was feeling…you know…such a head [brings both hands to the head with palms facing the temples and fingers wide]…You go to work willingly, you love it…you commit yourself…the thing is that you are tired of seeing the friend who avoids you, the colleague on the floor below who runs away…in short, you wonder why…how is it possible. (QR08)

I consider my colleagues them all friends, then suddenly you feel rejected, thrown away, outside…in short, the job is for everyone a second home…and they kicked me out of my house. (NM17)

Within the perceived stigma in the workplace, we found two subthemes: “gun-sight looks” and “avoiding closeness to others”. These signal experienced behaviours and the perception of attitudes that left a mark on the participants who probably did not expect such attitudes from work colleagues, professionals like them who had been trained to face delicate situations.

#### 3.1.1. “Gun-Sight Looks”

Even when the behaviour was not directly revealed, our participants perceived that they were the centre of particular attention. The feeling of being constantly judged, observed even from afar, was present in each of them. Feeling constantly observed made it difficult to enter the workplace. These episodes of perceived stigma fall under the classic definition of social stigma, right along clearly stigmatising episodes.

You see it in their eyes, how they look at you when you arrive or when you have to pass by them…their eyes are like gun sights…their impassive face and their fixed eyes [closes one eye as though looking through a gun sight] they follow your every move…maybe to see where you go, what you touch…in short, you always feel under scrutiny. (LI16)

You’re being constantly watched…it’s like going around with a sign on your back: ‘I harb our viruses and will infect you!’…you want to see that I really have the sign and that’s why… [mimics having a sign on her back, her head turned backwards in search of the sign]…in short, I walk and I am accompanied by silence and looks…fixed eyes… (OP07)

#### 3.1.2. “Avoiding Closeness to Others”

At other times, the stigma was manifested through precise and direct behaviours, initially underestimated by our participants. Accustomed to being an integral part of an inclusive community, they suddenly noticed these behaviours and classified them as clear acts of exclusion. The fact that they had always been an integral part of that environment likely clouded their judgment even before specific and clearly stigmatising behaviours.

At the beginning I didn’t care, then thinking about it…for example…the changing rooms are not shared, but the lift to get there, yes…and then you see that when you have to take it…well, it is only you…they start giving a thousand excuses, like, oh no, I forgot something in the car…or I am waiting for X colleague…but you know it is not so. Do they always forget something when they have to go to work? (EF03)

For a while, I honestly stopped going to the bar or the coffee machines to take anything…whereas before you could have a chat there, now you are almost avoided…you see that the face, as smiley as it has always been, darkens…the smile disappears…the gaze lowers…the colleague does not look at you and runs. (IL05)

### 3.2. Stigma in Everyday Life

All our participants highlighted how networks of friendships that had existed until a few days before had suddenly disappeared, almost as if they had vanished from normal social contact. The world of relationships that enveloped them and their loved ones appeared to have dissolved. The participants described well the feeling of perceived isolation. The description of how suddenly they found themselves in this situation, without having time to digest this change, was also clear. The descriptions of the behaviours that led to the isolation of the participants are clear.

A stain? No a stain comes off…I feel branded with fire…and I will not be able to forget…I always think about it, every day, every evening and it hurts. I feel like my colleague…they wrote in the lift that she was a carrier of COVID…to me they wrote it in a letter. Well I made as many copies as there is mailboxes in the condominium… and I hung them everywhere… they did not even have the courage to apologise [the face is red, the gaze fixed and stern, but the eyes appear shiny]…before this you saw that the doors closed when you came near, or the windows. At home I do the washing and when I hang my clothes I notice that the eyes are lowered, to avoid min. hearing those balcony doors slamming hurts. (FE14)

Let’s say that until a few months ago I was almost the mascot of the building…everyone was caring…since the beginning of the pandemic I speak with few people, almost no one…many have become elusive… and sadly, my rent contract may not be renewed…the explanation was that the condominiums are afraid of my work…they are afraid of being next to someone who works in the hospital. (DC13)

Within the stigma found in everyday life, we found two subthemes: “nobody wants to touch you” and “the fault of being your family members”.

#### 3.2.1. “Nobody Wants to Touch You”

Some episodes, after reflection, led our participants to important considerations. Objects were perceived as their continuation. The objects owned by our interviewees were seen as an integral part of them, considered to be dangerous because they belonged to a certain category of people.

In the garage where I park the car now they say hello to me from the cage, and nobody moves my car anymore… the caretaker also told me that I can take the keys home because nobody wants to touch it. (BA12)

The gas station invites you to do self-service…he knows that you work with infected people erects a wall between you and him…at the supermarket, where they have known me and my family for years now…the cashiers, not all of them though, do not look up from the products and even avoid touching your ATM card. (RQ19)

#### 3.2.2. “The Fault of Being Your Family”

Our participants also described how stigma was extended to even their dearest affections. The stigma seemed to be rapidly extended to the relatives of the nurses subjected to stigma, with people clearly avoiding not only nurses, but also their families, probably out of fear of the spread of infection. This influences the lived experiences of the nurses and also has implications for the mental health of family members who are subjected to stigma.

I can tolerate everything, the fear of me, of my job, but that they take it out on my children… I don’t accept it [dark and angry face, slams his fist on the table and turns up the volume, as if to be heard]. They are children… I do not have the possibility to stay in a hotel, I sleep alone, I eat alone, I see my children from afar, as well as my wife… and what are these people doing? Weren’t they let them in the elevator? They closed the door in your face. And it’s not the first time… when the oldest, who is 15, goes down to buy something, the doorman avoids him and some people ask them to hurry up and not to stay in the building. (UV10)

My wife now can’t even get her shopping carried…at the beginning I said to her ‘but no, you’re wrong…it must be your impression’…then it happened to me…her fault is being my wife…and you know why…because those who did not know, like the butcher of the supermarket, once he found out, he too began to avoid talking to her, serving her. (PO18)

## 4. Discussion

This study investigated the experience of stigma perceived by nurses during the COVID-19 pandemic. Understanding this stigma, how it can be involuntarily generated, and how it is perceived, are fundamental to protecting health workers’ wellbeing and thereby the wellbeing of the general population during epidemics and pandemics. The perception of stigma by these workers can influence their mental health [16,29,47] and their work. During an epidemic, people adopt prejudicial behaviours, consciously and unknowingly, in response to their fear of getting sick [30]. Fear, anxiety, negative attitudes, neglecting behaviour, and rejection were found to contribute to episodes of stigma, which may affect the mental health of healthcare professionals [16,29,47] in both the short [16] and long [48] term. During previous epidemics, health workers were almost systematically stigmatised because of their work [28,29]. Our results show that stigma is a very frequent phenomenon in this population. Two main themes that shaped their experiences were identified: the area of working life and that of everyday social life.

The first themes revolved around the experience of exclusion and social prejudice in the workplace, namely, the hospital, by colleagues. Analysing our data showed that, just as described in the literature [49], episodes of isolation and discrimination within the working context occurred with regard to nurses working in COVID-19. In the health sector, stigma within the workplace is essentially found in those who actively provide assistance to infected subjects [50]. The detailed analysis of our results showed a first subtheme characterised by a particular form of social exclusion by colleagues. Participants felt that they were under surveillance, subjects being keenly observed from a distance, like in “gunsights”, as described by some of them. People’s desire to avoid infection and remain healthy apparently produced, subtly or unclearly, social rejection behaviours towards their colleagues working in COVID-19 departments. This form of discrimination, and the experience of evasion and exclusion from colleagues induced changes in habitual behaviours, such as the avoidance of bars or public areas.

The second subtheme that emerged revealed participants’ experiences linked to their colleagues’ emotional and social distancing. For our participants, avoidance and averted or lowered glances by colleagues characterised the process of stigmatisation and connote the clear separation between “us” and “them” [51].

It is uncommon to find such a detailed and thorough description. The literature instead shows how emotional support from colleagues in similar situations is important [8,18,29]. Our results show how incidents of stigmatisation are quite common in the daily and domestic life of health personnel. This social discrimination influenced their daily activities, and in specific cases, even those of their family members. Participants described the limitations imposed on their habits; the repeated behaviours towards them limiting almost any form of community property use, such as public transport or a condominium lift [52]. During the SARS epidemic in the early 2000s, similar incidents in Singapore were described. Taxi drivers were refusing to transport health workers, just as health workers were forbidden to travel on the subway in uniforms [29]. Participants said that they had suffered severe discriminatory behaviour from people that they knew in their daily lives, such as their building’s porters, and cashiers at their grocery store or gas station, simply for being nurses working in hospitals. Concern about possible infection led to friends becoming distant, as reported by health professionals in previous experiences [29].

Participants perfectly conveyed the idea of how such behaviours can leave their mark [53]. Unwillingness to touch material items owned by the stigmatised nurses [48] has greatly contributed to the perceived social isolation. As shown in previous scientific studies, these behaviours are also extended to the nurses’ families [16,19,52,54]: Korean nurses who worked in hospitals with MERS-CoV patients were turned away from friends and sometimes family members, they were forbidden to use the elevators where they lived, and their children were not allowed to attend kindergartens and schools (Jung and Cho, 2015). This form of stigmatisation of family members is also evident from our data. In some cases, the participants experienced further isolation.

Understanding this experience of stigma can lay the foundations for the creation of specific intervention programmes aimed at protecting the health of these professionals and the general population.

This study has some limitations. The sample comprises predominantly nurses operating in a single territorial area. Any social and cultural differences within the national territory may not be included. The sample size, obtained when the data are saturated, cannot be considered to be a limitation for the type of study. Our results serve as a starting point for future studies on both the nursing and other populations, such as physicians, who are generally less susceptible to stigmatisation.

## 5. Conclusions

Public health emergencies, such as the COVID-19 pandemic, are stressful events for individuals and communities. Fear and anxiety about an illness can lead to social stigma towards people, places, or things. Stigma, fear, and uncertainty can act as barriers to appropriate medical and mental health interventions [55]. Stigma is more dangerous than the disease itself, as stated by the WHO.

Given the history of infectious diseases, more are likely to emerge in the times to come, which would present new challenges for which we must be ready. It is essential, from the early organisational stages in epidemic prevention, to provide healthcare professionals involved in direct care with a professional and flexible psychological intervention [8,56,57].

Joint efforts to reduce stigmatisation about COVID-19 must be carried out jointly by the most prominent exponents of various sectors of our society. Furthermore, only and exclusively reliable and official sources of information should be promoted, and immediate support inside and outside the workplace should be provided to stigmatised individuals.

## Figures and Tables

**Table 1 healthcare-10-00025-t001:** Socio-demographic characteristics of nurses (n = 19).

Identifier	Sex	Age	Marital Status	Kids	Working Region	Ward/Work Area	COVID Ward	Years of Service
AB01	M	42	In a relationship	1	Campania	Subintensive COVID	YES	16
CD02	F	44	Married	2	Campania	UTIC	NO	20
EF03	F	48	Married	2	Campania	Respiratoryrehabilitation	YES	18
GH04	M	60	Married	3	Campania	Urology	NO	39
IL05	M	38	In a relationship	0	Campania	T. subintensive	YES	15
MN06	F	27	Single	0	Campania	Radiology	*	2
OP07	M	25	Single	0	Piedmont	Resuscitation	YES	2
QR08	F	29	In a relationship	0	Campania	Resuscitation	YES	5
ST09	F	39	Married	3	Marche	Operating room	NO	14
UV10	M	44	Married	2	Campania	T.subintensive	YES	21
ZZ11	M	46	In a relationship	0	Campania	Neurosurgery	NO	23
BA12	M	50	Married	1	Campania	ORL	NO	27
DC13	M	29	Single	0	Tuscany	Intensive therapy	YES	5
FE14	F	39	In a relationship	0	Lombardy	Cardiology	NO	11
HG15	F	44	Married	2	Campania	Gynaecology	NO	18
LI16	F	41	Married	2	Campania	Transport	*	14
NM17	F	44	Married	3	Lazio	Internal Medicine	NO	16
PO18	M	47	Married	2	Lazio	T.subintensive	YES	11
RQ19	M	54	Married	3	Campania	Haematology	NO	28

* service for all (COVID− and COVID+ subjects).

## Data Availability

Not applicable.

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
