# Peer review of "Nurses and Stigma at the Time of COVID-19: A Phenomenological Study"

_healthcare, 2021, doi:10.3390/healthcare10010025_

Round 1

Reviewer 1 Report

Thank you very much for offering this opportunity to review this manuscript. I enjoyed reading it. The purpose of this study is “to explore the lived experience of Italian nurses about perceived 14 stigma during COVID-19 pandemic.” This paper is well structured and concise. The manuscript is generally well written; reflects rigorous methodology; and contributes to the understanding how healthcare professionals experience stigma, which is essential to design and implement specific educational, psychological and organizational programmes.

My only comment is that I wonder if it is possible for the authors to clearly propose the suggestions or recommendations on how to protect the health of these professionals and also of the general population, which may reinforce the significance of the study.

Author Response

Comments and Suggestions for Authors

Thank you very much for offering this opportunity to review this manuscript. I enjoyed reading it. The purpose of this study is “to explore the lived experience of Italian nurses about perceived 14 stigma during COVID-19 pandemic.” This paper is well structured and concise. The manuscript is generally well written; reflects rigorous methodology; and contributes to the understanding how healthcare professionals experience stigma, which is essential to design and implement specific educational, psychological and organizational programmes.

Thanks for taking the time to evaluate our manuscript. We are confident that following your suggestions the manuscript will have greater strength

My only comment is that I wonder if it is possible for the authors to clearly propose the suggestions or recommendations on how to protect the health of these professionals and also of the general population, which may reinforce the significance of the study

Thank you to your valuable suggestion, we have now briefly tried to clarify in the "conclusions" section which interventions we consider feasible in the immediate future and useful. In the new version of the manuscript you will be able to read “Joint efforts to reduce the stigmatization about COVID-19 must be carried out jointly by the most prominent exponents of the various sectors of our society. Furthermore, only and exclusively reliable and official sources of information should be promoted, and immediate support, inside and outside the workplace, should be provided to stigmatized categories.”

Reviewer 2 Report

This is an interesting article that explores Italian nurses experience of stigma. There are significant weaknesses in the description of methods and the language is often ungrammatical and confusing.

Methods

The description of the sample is inadequate. There is no discussion of the sampling frame; how were potential participants contacted. For instance did all the nurses work in hospital settings or work in the same city or region? This is important as work contexts shape the experience of stigma and difference is locality are likely to be linked to different degrees of infection.

The account of the research design is inadequate. There is no description of the design of the study but simply a reference to a single text book.

How many of the authors carried out the interviews; the text simply says ‘researcher’ or ‘researchers’ (lines 113 and 120). Later in the discussion of analysis the text says ‘The data analysis was performed by each researcher…” (lines 131-132). It is never made clear how many of the 6 authors conducted the interviews or undertook the analysis. How was ‘saturation’ judged? Was the analysis conducted simultaneously with the collection of the data? If not then the decision to stop after 19 interviews was not based on ‘data saturation’ (line 118).

There is little difference between the Results and the Discussion section. The latter is just a reiteration of the former without recourse to any theoretical concept to provide an interpretation. The result, while interesting, is simply a description of the experience of 19 nurses.

Sources from the World Health Organisation should not be cited as Organisation, World Health.

Overall, while the data are potentially interesting significant additional work is needed before this article is of publishable quality.

Author Response

Comments and Suggestions for Authors

This is an interesting article that explores Italian nurses experience of stigma. There are significant weaknesses in the description of methods and the language is often ungrammatical and confusing.

Thanks for the valuable tips provided. We are confident that they will help to give our manuscript greater strength

Methods

The description of the sample is inadequate. There is no discussion of the sampling frame; how were potential participants contacted. For instance did all the nurses work in hospital settings or work in the same city or region? This is important as work contexts shape the experience of stigma and difference is locality are likely to be linked to different degrees of infection.

Thanks for the valuable suggestion. We have now proceeded to better specify the characteristics of our sample here and in the results section

 The account of the research design is inadequate. There is no description of the design of the study but simply a reference to a single text book.

Thank you for paying attention to this aspect, We have now provided a better description.. Inside the new manuscript you will now be able to read: Descriptive phenomenology requires the suspension of all ideas and suppositions in order to access the nature of consciousness itself, in this way it will be possible to obtain the meaning of the individual's experience. In descriptive phenomenology, daily conscious experiences must be described while any preconceived opinions must be set aside and placed in parentheses (Creswell J,1994).

Interpretative phenomenology is an approach that aims to provide detailed examinations of personal lived experiences. It is a particularly useful methodology for examining complex, ambiguous and emotionally charged topics (Smith JA, et al, 2015)

Cohen's phenomenology (Cohen, 2000) is divided into the following phases: (1) bracketing: critical reflection technique that precedes data collection. Researchers are asked to write their own beliefs and hypotheses on the phenomenon to avoid contamination of the analysis by prejudices on the subject; (2) interview of the participants with open questions. During the interviews the researcher will take notes in the field, noting the non-verbal language of the body, describing the environment and any noteworthy detail. The interviews will continue until the data is saturated; (3) Interviews will be transcribed verbatim and will be checked for accuracy; (4) investigators will immerse themselves in the data by repeatedly reading and rereading interview transcripts and field notes; (5) extrapolation of the main themes and any sub-themes; (6) the extrapolated themes are compared among all the researchers involved, discussing and determining their reliability (7) the extracted themes are presented to the participants of the study participants for their confirmation; (8) write the scientific report .

 How many of the authors carried out the interviews; the text simply says ‘researcher’ or ‘researchers’ (lines 113 and 120). Later in the discussion of analysis the text says ‘The data analysis was performed by each researcher…” (lines 131-132). It is never made clear how many of the 6 authors conducted the interviews or undertook the analysis.

We have now clarified this aspect by indicating the 3 authors who conducted the interviews, all 6 authors analyzed the data

 How was ‘saturation’ judged? Was the analysis conducted simultaneously with the collection of the data? If not then the decision to stop after 19 interviews was not based on ‘data saturation’ (line 118).

We apologize for this lack of clarity. Replying to the previous comment about methods design we have tried to clarify this aspect

There is little difference between the Results and the Discussion section. The latter is just a reiteration of the former without recourse to any theoretical concept to provide an interpretation. The result, while interesting, is simply a description of the experience of 19 nurses.

Sources from the World Health Organisation should not be cited as Organisation, World Health.

 Done

Overall, while the data are potentially interesting significant additional work is needed before this article is of publishable quality

We hope the new version of manuscript has improved in the shortcomings highlighted

Reviewer 3 Report

The subject of this review is the paper entitled "Nurses and Stigma at the Time of Covid-19. A Phenomenological Study". The aim of the study was to assess the impact of the Covid-19 pandemic on the stigmatization of medical personnel, with particular emphasis on nurses, and the further consequences of this stigma. I find this topic interesting and it is necessary to talk about it openly, because, as Authors also indicate, patients with COVID-19 are not the only one who suffers because of the pandemia, but healthcare workers also experience many difficulties and mental and physical stress.
Despite the importance of the topic, mistakes and errors within the text are noticeable.

Authors should carrefully read their manuscript because it is full of missing spaces which makes text difficult to read. Below I listed other mistakes that should be corrected before publication.

Abstract

The headings within the abstract ("background", "methods" ...) should be deleted, according to "instructions for authors" at Healthcare journal.

Line 10: "The COVID-19 pandemic is putting strain on the entire planet and its health systems" -This sentence needs to be corrected because of its poor logic. Now the Authors suggest that planet (as an astronomical object) hurts because of the pandemic as well as ITS health systems. I would suggest to re-write this sentence into, for example, "The COVID-19 pandemic is putting strain on every country in the world and their health systems"

Line 14: "To  explore  the  lived  experience  of  Italian  nurses  about  perceived  stigma  during COVID-19 pandemic" - this sentence looks like it's out of context. I suggest to correct it for example inserting "the aim of this study"

Instroduction

A lots of spaces between words and/or puntuation marks are missing. The introduction section looks like being written in a hurry. It does not look good and makes the text much more difficult to read. Please read carrefully the whole text (introduction and the other parts) and correct all those mistakes. Below some examples of missing spaces:

Line 28: space is missing (...SARS-CoV-2which...)

Line 34-35: space is missing (...transmission(Sun et al., 2020)and... )

Line 41: space is missing (...Hemalatha, 2020).Older...)

Line 42: space is missing (...COVID-19(Abd El-Aziz & Stockand, 2020)...)

Line 50: Correct COVID 19 to COVID-19.

Material and methods

Line 89, 2.2. Sample. I don't feel that calling people as "sample" is correct. I suggest to use phrase "tested group" like for example, instead of sentence line 152 "our sample consisted of 19 nurses" use "Tested group consisted of 19 nurses" or "We examined 19 nurses".

Line 99-100: "each investigator involved in the study" does it mean in this study, there was more than one investigator/interviewer? How it could have influenced the obtained results?

Line 117: what is that "non-verba578l"? Looks like typo bug that needs to be corrected.

Results

Tested group is very small. 19 nurses seem to be an insufficient group to draw conclusions relating to the entire large group of medical workers, not only in Italy but also around the world. I suggest to call the entire study as a pilot study. This would give a good introduction to further analyzes on a larger group of people, also multi-center. Therefore I suggest to change the title of the work to "Nurses and stigma at the time of Covid-19. A phenomenological, pilot study"

Line 152: DS is a shortcut for standard deviation? If yes, it should be SD not DS.

Discussion

Line 281: Authors cannot use the phrase "stigma perceived by nurses during the  COVID-19 pandemic." based on 19 nurses only. Authors should not generalized having so small tested group, therefore I strongly suggest to revise the whole study and treat it like a pilot study.

Line 345: Adding information about the limitation of the study is very valuable and demonstrates the integrity of the researchers. The suggested change of the topic of work with the annotation "pilot study" will only complete this context. 

Reference list has to be changed according to the "instructions for authors": (In the text, reference numbers should be placed in square brackets [ ], and placed before the punctuation; for example [1], [1–3] or [1,3]. )